# Adherence and sustainability of interventions informing optimal control against the COVID-19 pandemic

Laura Di Domenico[1,6], Chiara E. Sabbatini[1,6], Pierre-Yves Boëlle[1], Chiara Poletto[1], Pascal Crépey [2], Juliette Paireau[3,4], Simon Cauchemez[3], François Beck[4], Harold Noel [4], Daniel Lévy-Bruhl[4] & Vittoria Colizza[1,5✉]

## Abstract

**Background** After one year of stop-and-go COVID-19 mitigation, in the spring of 2021 European countries still experienced sustained viral circulation due to the Alpha variant. As the prospect of entering a new pandemic phase through vaccination was drawing closer, a key challenge remained on how to balance the efficacy of long-lasting interventions and their impact on the quality of life.

**Methods** Focusing on the third wave in France during spring 2021, we simulate intervention scenarios of varying intensity and duration, with potential waning of adherence over time, based on past mobility data and modeling estimates. We identify optimal strategies by balancing efficacy of interventions with a data-driven "distress" index, integrating intensity and duration of social distancing.

**Results** We show that moderate interventions would require a much longer time to achieve the same result as high intensity lockdowns, with the additional risk of deteriorating control as adherence wanes. Shorter strict lockdowns are largely more effective than longer moderate lockdowns, for similar intermediate distress and infringement on individual freedom.

**Conclusions** Our study shows that favoring milder interventions over more stringent short approaches on the basis of perceived acceptability could be detrimental in the long term, especially with waning adherence.

## Plain language summary

In the spring of 2021, social distancing measures were strengthened in France to control the third wave of COVID-19 cases. While such measures are needed to slow the spread of the virus, they have a significant impact on the population's quality of life. Here, we use mathematical modelling based on hospital admission data and behavioural and health data (including data on mobility, indicators of social distancing, risk perception, and mental health) to evaluate optimal COVID-19 control strategies. We look at the effects of interventions, their sustainability and the population's adherence to them over time. We find that shorter, more stringent measures are likely to have similar effects on viral circulation and healthcare burden to long-lasting, less stringent but less sustainable interventions. Our findings have implications for the design and implementation of public health measures to control future COVID-19 waves.

[1] INSERM, Sorbonne Université, Pierre Louis Institute of Epidemiology and Public Health, Paris, France. [2] Univ Rennes, EHESP, REPERES « Recherche en Pharmaco-Epidémiologie et Recours aux Soins »—EA 7449, 35043 Rennes, France. [3] Mathematical Modelling of Infectious Diseases Unit, Institut Pasteur, UMR2000, CNRS, Paris, France. [4] Santé Publique France, French National Public Health Agency, Saint-Maurice, France. [5] Tokyo Tech World Research Hub Initiative, Institute of Innovative Research, Tokyo Institute of Technology, Tokyo, Japan. [6] These authors contributed equally: Laura Di Domenico, Chiara E. Sabbatini. ✉email: vittoria.colizza@inserm.fr

The emergence of the SARS-CoV-2 Alpha variant in December 2020[1,2] disrupted the management of COVID-19 pandemic in Europe. The alert arrived as some governments were lifting interventions that had been applied to curb the second wave. Some countries, such as the UK and Ireland, were forced to rapidly implement strict lockdowns to control the explosion of cases due to the variant. Others maintained or strengthened their restrictions because of concerns over the new variant[3].

Few months after, with vaccination lagging behind (25% of the population of the European Union with a first dose on May 1, 2021 vs. 44% in the US, 51% in the UK, and 62% in Israel[4]) and a third wave due to the Alpha variant, continental Europe faced the challenge of relying once again on heavy restrictions to reduce sustained viral circulation and improve the epidemic situation approaching the summer. But what is the optimal strategy, given vaccination rollouts, the epidemic conditions, and the sustainability of long-lasting restrictive policies? On one side, limited available options remain beyond high intensity interventions, once milder layers of social distancing have been accumulated, strengthened, and extended over time (e.g., curfew, closure of restaurants and bars, closure of schools). On the other side, the efficacy and long-term sustainability of the adopted policies are potentially threatened by loss of adherence and policy-induced fatigue[5,6], affecting the quality of life of the population.

Building on observed adherence waning and introducing a data-driven measure capturing the limitations on individual freedom resulting from restrictions, we compared intervention scenarios of varying intensity and duration, and examined the role of adherence and sustainability on optimal epidemic control. The study is applied to the third wave in Île-de-France—the Paris region, the most populated of France and heavily hit by the pandemic—accounting for vaccination rollout plans, seasonality, and plans for the phasing out of restrictions.

We show that long-lasting interventions of moderate stringency achieve the same reductions in viral circulation and healthcare burden of shorter but higher stringency restriction, but at the expense of a higher distress in the population. This is exacerbated if adherence to policy wanes over time.

## Methods
### Data
*Hospital surveillance data.* We used regional daily hospital admission data, collected in the SIVIC database[7]. The database includes the number of admissions of COVID-19 confirmed patients to regular hospital or intensive care units. Hospital data are corrected for notification delays and do not suffer changes in detection or sampling, unlike the number of detected cases. As such, they provide a robust data source and have been used throughout 2020 in France for pandemic assessment and response[8–11].

*Mobility data.* Mobility reductions shown in Fig. 1 were extracted from two different data sources. Overall mobility was reconstructed from mobile phone data provided by Orange Business Service Flux Vision[12,13]. Data included origin-destination travel flows of mobile phone users among 1436 geographical areas in France. Each area corresponds to a group of municipalities, defined according to the 2018 EPCI level (Établissements Publics de Coopération Intercommunale[14]). Mobility reduction in a given week was computed as the relative variation of the number of trips with respect to the prepandemic baseline. Estimated presence at workplaces was obtained from Google Mobility Reports[15]. This dataset provides the relative change in the daily number of visitors to places of work compared to a prepandemic baseline, based on Google location-history data.

*Indicators of social distancing, risk perception, mental health.* Several initiatives collect data over time through surveys to explore individual behaviours in response to COVID-19 pandemic. Here we use data from YouGov[16] and Santé publique France[17]. Surveys gather self-reported data, tracking compliance with preventive measures (e.g., avoiding social gatherings or contacts with other people, frequency of the use of masks), as well as risk perception and mental health indicators (e.g., fear to contract the virus, anxiety, depression). Indicators for specific social distancing behaviors (avoiding gatherings, use of masks) are used in addition to mobility data described above. YouGov surveys cover multiple countries and provide data at least every 2 weeks. Santé publique France polls collect data at the national level at least every month.

*Ethics statement.* Orange Business Service Flux Vision aggregated mobility travel flows were previously anonymised in compliance with strict privacy requirements, presented to and audited by the French data protection authority (CNIL, Commission Nationale de l'Informatique et des Libertés). They were accessed under license for this study. The study did not require an ethical approval as it involved review of publicly available documents, involved analyses that were based on previously published studies, involved aggregated and anonymous data, did not involve evaluation of experimental or patient data.

**SARS-CoV-2 two-strain transmission model.** We used a stochastic discrete age-stratified two-strain transmission model, integrating data on demography[18], age profile[18], social contacts[19], mobility[15], and adoption of preventive measures[17]. The model accounts for the co-circulation of two strains and vaccination. Four age classes are considered: [0–11], [11–19], [19–65], and 65+ years old (children, adolescents, adults and seniors respectively). Transmission dynamics follows a compartmental scheme specific for COVID-19 (Supplementary Fig. 1) where individuals are divided into susceptible, exposed, infectious, hospitalized, and recovered. The infectious phase is divided into two steps: a prodromic phase ($I_p$) and a phase where individuals may remain either asymptomatic ($I_{as}$, with probability $p_a = 40\%$[20]) or develop symptoms. We distinguished between different degrees of severity of symptoms, ranging from pauci-symptomatic ($I_{ps}$), to individuals with mild symptoms ($I_{ms}$), or severe symptoms ($I_{ss}$) requiring hospitalization[11,21]. The duration of the infectious period was computed from the estimated mean generation time of 6.6 days[22] (Supplementary Fig. 2). Prodromic, asymptomatic and pauci-symptomatic individuals have a reduced transmissibility[23]. A reduced susceptibility is considered for children and adolescents, along with a reduced relative transmissibility for children, based on available evidence[24–27]. We assume that infectious individuals with severe symptoms reduce of 75% their number of contacts because of the illness they experience. Parameter values and corresponding sources are reported in the Supplementary Table 1. Sensitivity analysis on the probability of being asymptomatic, the susceptibility of younger age classes and transmissibility of children was performed in previous works[8,9,28].

Contact matrices are parametrized over time to account for behavioral response to social distancing interventions and adoption of preventive measures. Contacts at school, work and on transports are considered according to the French school calendar, school closures, and presence at workplaces estimated by Google. Physical contacts are reduced based on data from regular large-scale surveys conducted by Santé Publique France[8]. Contacts engaged by seniors are subject to an additional reduction of 30%,

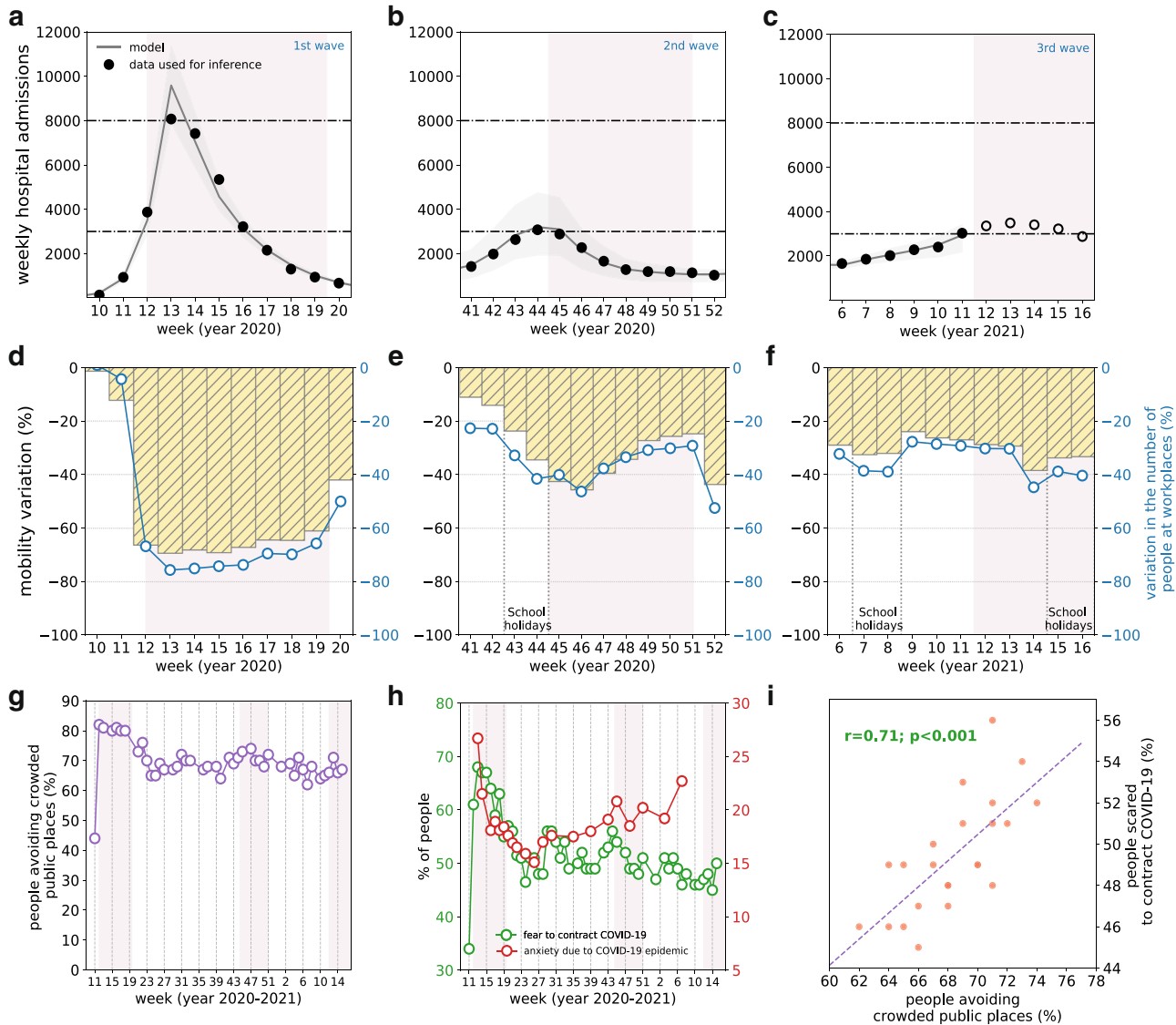

**Fig. 1 COVID-19 pandemic waves in Île-de-France, with associated mobility reductions, social distancing, risk perception, and psychosocial burden. a–c** Weekly hospital admissions in Île-de-France during the first (**a**; weeks 10–20, March 2–May 17, 2020), second (**b**; weeks 41–52, October 5–December 27, 2020), and third (**c**; weeks 6–16, February 8–April 25, 2021) pandemic wave. Dots refer to data; filled dots correspond to the data used to fit the model, void dots correspond to data outside the inference window. Curves and shaded areas correspond to median fitted trajectories and 95% probability ranges, obtained from n = 250 independent stochastic runs. Horizontal dashed lines refer to the peak of the first and second wave in the region. **d–f** Mobility reduction in Île-de-France during the first (**d**), second (**e**), and third (**f**) pandemic wave. Yellow histograms represent the variation of mobility with respect to prepandemic levels, based on the number of trips extracted from mobile phone data[12]. Blue curves show the estimated change in presence at workplace locations over time with respect to prepandemic levels based on Google location-history data[15]. Shaded rectangles in the plots of the first two rows correspond to social distancing measures (strict lockdown in the first wave, moderate lockdown in the second wave, strengthened measures in the third wave). The second week of the second lockdown and the third week of the strengthened measures against the third wave have lower mobility and presence at workplaces due to bank holidays in the week. Vertical dotted gray lines correspond to school holiday periods. **g–i** Percentage of individuals avoiding crowded public places[16] (**g**), percentage of individuals scared to contract COVID-19[16] and prevalence of anxiety in the context of COVID-19 epidemic (**h**)[17] as functions of time; scattered plot between the percentage of individuals scared to contract COVID-19 and the percentage of individuals avoiding crowded places (**i**) in the time period October 2020–April 2021 (full time period shown in Supplementary Fig. S5), with the results of a Pearson correlation test (effect size 0.71, p-value < 10⁻³). Results for these indicators refer to the national scale. Shaded rectangles in panels **g**, **h** correspond to social distancing measures as in panels **a–f**.

to account for evidence of a higher risk aversion behavior of the older age class compared to other age classes[8].

**Alpha variant**. Genomic and virological surveillance to identify specific mutations are in place in France since the start of 2021 to monitor variants over time. The first large-scale genome

sequencing initiative (called Flash1 survey) was conducted on January 7–8 and analyzed all positive samples provided by participating laboratories[29]. The proportion of the Alpha variant in Île-de-France was estimated to be 6.9%, compared to the national estimate of 3.3%, making Île-de-France the region with the highest penetration registered in the country. Flash surveys are performed on average every two weeks on a sample of sequences.

Starting week 6, 2021 a new protocol for virological surveillance was implemented to provide more timely estimates on the weekly frequency of detected viruses with specific mutations. It was based on second-line RT-PCR tests with specific primers that allow the detection of the main mutations that characterize the variants of concern. They must include at least the N501Y mutation and allow to distinguish the Alpha variant from the Beta or the Gamma variants. The frequency of the Alpha variant over time in Île-de-France is reported in the Supplementary Fig. 3.

We considered the co-circulation of the Alpha variant together with the historical strains, assuming complete cross-immunity. An increase in transmissibility of 59% (95% credible interval: 54–65%)[29] was considered for the Alpha variant compared to the historical strains. This early estimate was obtained from the Flash1 and Flash2 survey in France, and it is in line with other estimates[1,2]. To account for uncertainty in the transmission advantage and possible changes due to restrictions, we also show for sensitivity the results assuming 40% of increase in transmissibility, i.e., the lower estimate provided by ref. [2]. (Supplementary Fig. 11). We considered a 64% increase in hospitalization rates, following evidence of an increased risk of hospitalization after infection due to the Alpha variant compared with other lineages[30,31]. The frequency of the Alpha variant was initialized in the model on January 7, 2021 using the estimates of the first large-scale nationwide genomic surveillance survey (Flash1). The model was validated against virological and genomic surveillance data[10] on prevalence of Alpha variant over time. The Alpha variant was estimated to become dominant in the region by mid-February 2021[10] (Supplementary Fig. 3).

**Vaccination rollout campaign**. Administration of vaccines was included in the model according to the vaccination rhythm adopted in France starting January 2021. We considered the administration of 100,000 doses per day (including first and second doses) at the national level from the end of January (w04), accelerated to 200,000 first doses per day starting the beginning of March (w10), and 300,000 first doses per day starting April (w13). Rollout plans were expressed in terms of first administrations from March on to follow the objectives of authorities, delaying the administration of the second dose to reach a higher coverage in a smaller timeframe. Higher vaccination paces (400,000–800,000 doses/day) were also tested (Supplementary Fig. 9). Paces are defined at the national level, and the number of doses is proportionally distributed to the region according to the population eligible for the vaccine. Vaccination is prioritized to the older age class, assuming 80% coverage, and then shifted to adults considering 50% coverage, according to surveys on vaccine hesitancy[32]. Vaccination to healthcare personnel and patients in long-term care facilities, performed at the start of the vaccination program, could not be explicitly included.

We considered 75% vaccine efficacy against infection[33] and 65% vaccine efficacy against transmission[34], estimated after the first injection. We further considered 80% vaccine efficacy against symptoms given infection, computed from the estimated vaccine reduction of symptomatic disease[34,35] estimated at 95% after the second dose, and found to be similar after the first dose[36]. As the landscape for vaccine efficacy rapidly evolves, we also tested vaccine efficacy against transmission equal to 40%[37] (Supplementary Fig. 13). We assumed efficacy to start 3 weeks after the first injection, and tested a delay of 2 weeks for sensitivity (Supplementary Fig. 14).

**Inference framework**. The model is fitted to daily hospital admission data through a maximum likelihood procedure, by estimating the transmission rate in each pandemic phase. More precisely, prior to the first lockdown and in absence of intervention (period January–March 2020), we estimated $\{\beta, t_0\}$ where $\beta$ is the transmission rate per contact and $t_0$ is the date of the start of the simulation. Then, in each phase we estimated $\alpha_{phase}$, i.e., the scaling factor of the transmission rate per contact specific to the pandemic phase under study (e.g., lockdown, exit from lockdown, summer, start of second wave, second lockdown, etc.). The transmission rate per contact in each phase is then defined as the transmission rate per contact in the pre-lockdown phase $\beta$ multiplied by the scaling factor $\alpha_{phase}$. A pandemic phase is defined by the interventions implemented (e.g., lockdown, curfew, and other restrictions) and activity of the population (school holidays, summer holidays, etc.). The effective reproductive number is derived from the estimated transmission rate through the next generation matrix approach[38]. The likelihood function is of the form

$$L(Data|\Theta) = \prod_{t=t_1}^{t_n} Poiss(H_{obs}(t)|H_{pred}(t,\Theta)) \qquad (1)$$

where $\Theta$ indicates the set of parameters to be estimated, $H_{obs}(t)$ is the observed number of hospital admissions on day $t$, $H_{pred}(t, \Theta)$ is the number of hospital admissions predicted by the model on day $t$ using parameter values $\Theta$, $Poiss(\cdot|H_{pred}(t, \Theta))$ is the probability mass function of a Poisson distribution with mean $H_{pred}(t, \Theta)$, and $[t_1, t_n]$ is the time window considered for the fit.

For Île-de-France, we seeded the model with 140 infected individuals to reduce the strong fluctuations associated with fitting the rapid increase and the high peak of hospitalizations observed in the first wave (the region was one of the areas mostly affected by the epidemic in early 2020). Simulations progress throughout 2020 to build immunity in the population. The model was validated against the estimates of three independent serological surveys conducted in France[8]. We used 250 stochastic simulations to compute median values and associated 95% probability range for all quantities of interest.

**First lockdown, second lockdown, curfew**. French authorities implemented two national lockdowns in 2020 to face the rapid surge of COVID-19 cases observed in the first and second wave. The first lockdown started on March 17, 2020 and lasted 8 weeks. It involved strict mobility restrictions outside home, together with closure of schools and non-essential activities. A less stringent lockdown was implemented for 6 weeks, starting on October 30, 2020. Schools remained open and a larger number of job sectors were allowed to operate. Measures were relaxed in the last two weeks of the lockdown, with the reopening of all retail for Christmas shopping. The second lockdown was lifted in mid-December with the application of a curfew starting at 8 pm, then anticipated in January 2021 to 6 pm to face increasing SARS-CoV-2 spread. Starting March 20, 2021, strengthened measures were additionally put in place in the region of Île-de-France to curb the third wave. These measures included mobility restrictions for trips exceeding 10 km, closure of business and of schools (1 week for primary schools, 2 weeks for middle and high schools in addition to 2-week school holidays in April). Values of the stringency index according to the timeline of interventions applied in France can be found in the Supplementary Fig. 4.

**Loss of adherence**. We used mobility data during the second lockdown and estimates of mobility reductions over time to assess if adherence to adopted policy waned over time, given unchanged restrictions. Focusing on the second lockdown, we compared the mobility reduction and reproductive number estimated in the first 3 weeks of lockdown implementation (w45–47, November 2–November 22, 2020) with respect to the following week. We

considered the average over the first-3-week period to smooth out the effect of the national holiday on November 11, altering mobility and presence at work with respect to a regular week.

**Lockdown scenarios**. Starting from week 12, 2021 (March 22, 2021), we compared a scenario assuming unchanged curfew conditions—as estimated in week 11 (*curfew scenario*)—with the trajectories resulting from the application of a lockdown for a duration of 2–8 weeks. We modeled the effect of a strict lockdown and a moderate lockdown based on measured mobility reductions and estimated transmissibility conditions during the first and second lockdowns, respectively, before relaxation emerged. The delay from the date of implementation of lockdown and the peak of hospitalizations was estimated to be 9 days during the first lockdown in the region, and varied between 7 and 12 days across regions[8]. In our scenarios we assumed a 7-day delay, and tested 10 days for sensitivity (Supplementary Fig. 12). We also tested lockdown scenarios with different starting dates, ranging from w11 to w15, 2020 (Supplementary Figs. 6-S7).

For lockdowns longer than 2 weeks, we compared scenarios assuming full adherence with situations characterized by a loss of adherence over time. We modeled the loss of adherence throughout interventions by a relative increase in the reproductive number, according to estimates from the second lockdown. We applied it after 2 weeks from implementation of interventions (to model a faster dynamics of adherence waning compared to the one observed in the second lockdown), and considered it limited in time (one drop) or continuous (repeated drops every two weeks).

**Distress index**. In order to quantify the infringement on individual freedom associated with lockdowns and provide a measure of the policy impact on the quality of life, we introduced a quantity called distress index. This measure takes into account both the duration and the intensity of restrictions. It is defined as the sum of the absolute values of weekly mobility reductions, over the number of weeks in which each restriction is maintained, and normalized to a scale from 0 to 10 (10 representing a strict 8-weeks lockdown and 0 the absence of restrictions). In case of a strict or moderate lockdown without loss of adherence, we considered the mobility reductions recorded during the two interventions in 2020, respectively, and varied durations from 2 to 8 weeks. Loss of adherence is computed with a variation of the mobility reduction after 2 weeks (limited loss) and repeated every 2 weeks (continuous loss), according to estimates from the second lockdown. We took the end of January 2021 (w04) as reference for the mobility reduction associated with curfew.

**Seasonality**. Multiple studies have investigated the relationship between SARS-CoV-2 transmission and weather factors, including temperature, humidity, ultraviolet radiation[39], suggesting that summer conditions may help in reducing transmission of the virus. Seasonal factors and simultaneous social distancing interventions are difficult to disentangle; however, containment measures are estimated to have a larger impact on the epidemic compared to seasonal effects only. Considering the estimated dependence of the reproductive number on UV radiation[40] and temperature[41], we extracted data on downward UV radiation at the surface and daily temperature recorded in Paris, in Île-de-France, in the last three years (2018–2020)[42] to derive an approximate estimate of the reduction in the transmission rate induced by climate factors for the region under study.

**Reporting summary**. Further information on research design is available in the Nature Research Reporting Summary linked to this article.

## Results

### Adherence and impact of interventions of varying stringency and duration.

During the strict lockdown implemented to curb the first wave (March–May 2020), mobility showed a reduction of 68.9% in the region compared to the prepandemic level (65% reduction at the national level[12]) that remained fairly constant over time (Fig. 1a, d). The associated effective reproductive number was estimated to be 0.73 [95% confidence interval: 0.72, 0.74]. During the second wave (October–December 2020, Fig. 1b), a less stringent lockdown was enforced, corresponding to an effective reproductive number of 0.88 [95% CI: 0.86, 0.90] estimated in the first 3 weeks of implementation (w45–47, November 4–22, 2020), before relaxation occurred. Recorded mobility and estimated presence at work decreased but remained almost two times higher compared to the first lockdown (average mobility reduction of 42.6% in the first 3 weeks compared to prepandemic levels) and showed a rapid and marked increase over time (Fig. 1e). This loss of adherence occurred remarkably faster (after the third week) and more substantially during the second lockdown compared to the first. The mobility reduction with respect to the prepandemic phase went from 42.6% in the first 3 weeks to 34.3% in the fourth week of the lockdown (w48, November 23–29, 2020), corresponding to a relative change of 19%. This was associated to an estimated relative increase of 10.9% in the effective reproductive number. Higher mobility was registered later, in the last 2 weeks of the lockdown (w49–50, November 30–December 13, 2020), due to the reopening of shops. The second lockdown was lifted with the application of an 8 pm curfew, then anticipated to 6 pm in January. The resulting effective reproductive number was estimated to be 0.90 [95% CI: 0.86–0.93] for the historical strains and 1.43 [95% CI: 1.37–1.48] for the Alpha variant at the end of January[10].

Indicators obtained from surveys report that fear of contracting COVID-19 showed an overall decrease over time after the second wave in France, whereas prevalence of anxiety in the population showed an increasing tendency, despite the lower stringency of restrictions. Performing a linear regression in this time window (i.e., October 2020–April 2021), we found a weekly average reduction of $-0.31\%$ for individuals scared to contract the virus, and $-0.39\%$ for individuals avoiding crowded places. In the same time window, we found an average weekly increase of $+0.13\%$ in the prevalence of anxiety in the population (Fig. 1h). Fear of contracting COVID-19 showed a positive correlation with the behavior of avoiding crowded places (Pearson $r = 0.71$, $p < 10^{-4}$, in the time period from w40 (September 28–October 4, 2020) to w15 (April 12–18, 2021), shown in Fig. 1g; results are robust when extending the timeframe of analysis). We observed a non-significant association between the prevalence of anxiety in the population and adoption of social distancing (Pearson $r = 0.2$, $p = 0.46$, in the time period from w11 in 2020 (March 9–15, 2020) to w11 in 2021 (March 15–21, 2021) (Supplementary Fig. 5).

Starting mid-February 2021, the region witnessed a sustained rise in hospitalizations leading to the start of the third wave (Fig. 1c). We fitted the model up to week 11 (March 15–21, 2021), when the region was still under curfew before strengthened measures were applied on March 20 to control the third wave. We then simulated intervention scenarios starting week 12, 2021, with stringency, efficacy and potential loss of adherence informed by past mobility data[12] and modeling estimates[8,9].

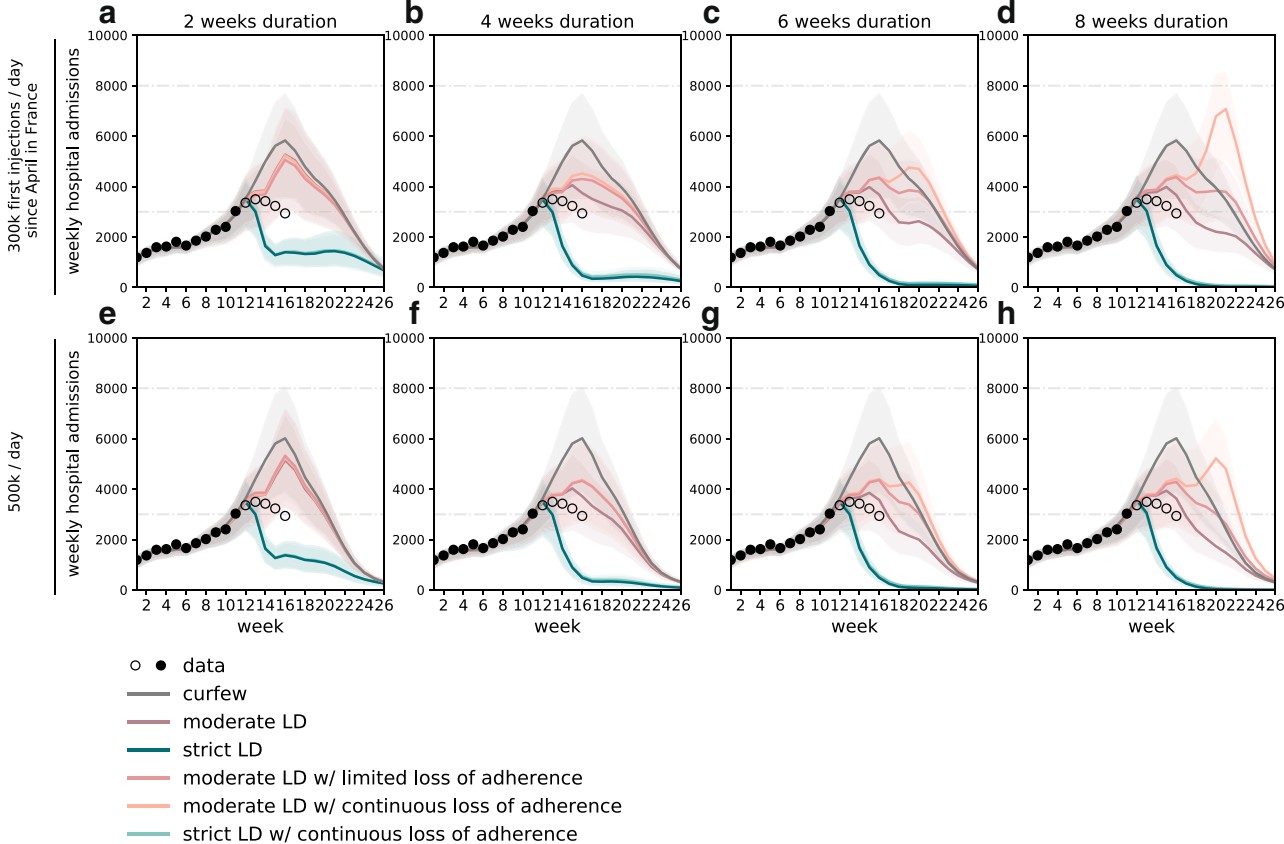

**Fig. 2 Timecourse of weekly hospital admissions in Île-de-France for lockdown scenarios of varying stringency, duration, and adherence. a–d**: vaccination pace accelerated to 300,000 first doses/day since the start of April, lockdown duration of two (**a**), four (**b**), six (**c**), or eight (**d**) weeks.; **e–h**: as in **a–d**, but assuming 500,000 first doses/day. Interventions are applied in w12 and assume a delay of one week to the peak in hospital admissions. Dots refer to data; filled dots correspond to the data used to fit the model and to provide the trajectory for the curfew scenario; void dots correspond to more recent data. Curves refer to the median trajectory; shaded areas around the curves correspond to the 95% probability ranges obtained from $n = 250$ independent stochastic simulations. The type of intervention is coded by different line colors; the abbreviation LD stands for lockdown. Horizontal dashed lines refer to the peak of the first and second wave in the region. Results for 2 weeks lockdown scenarios with or without loss of adherence overlap, as loss of adherence occurs in the third week. Results for strict lockdown scenarios with full adherence or loss of adherence overlap; for this reason, we do not show the scenario with limited loss of adherence.

Regardless of adherence, the strict lockdown was predicted to be the only measure able to achieve a rapid decrease of the epidemic trajectories (Figs. 2 and 3), in line with observations in the UK and Ireland following similar interventions. It would outperform moderate lockdowns of any duration, on both short- and longer-term epidemic impacts. Starting from about 3000 weekly hospitalizations at the time measures were applied, admissions would be reduced to less than 400 when exiting a strict lockdown of at least 1 month vs. more than 2000 after a moderate lockdown (Fig. 3a). Even with adherence waning, a strict lockdown would reduce the epidemic to the levels recorded at the exit of the first lockdown in May 2020 (670 weekly admissions in w20, 433 in w21, 330 in w22 in 2020), and it would be maintained low by increasing immunization rates. These levels would also enable a better control of viral circulation through test-trace-isolate when partially alleviating restrictions[8,43]. Importantly, a short circuit-breaker[44] of 2 weeks, after which curfew was restored, was predicted to be already enough to rapidly reduce hospitalizations to levels below the ones of February 2021. Rebounds at the end of the short lockdown would be prevented by maintaining a certain degree of social distancing (curfew) and increasing immunization, with stronger reductions over time for increasing vaccination rhythms (from 300,000 to 500,000 first doses/day since April; Fig. 2).

Obtaining results equivalent to a short strict lockdown would require moderate interventions to last longer than 2 months, and could potentially be compromised by loss of adherence to restrictions (Fig. 3a). This could slowdown and stop the decrease in hospital admissions, leading to a plateau or a rise in hospitalizations after several weeks of moderate lockdown, potentially higher than the peak of the third wave (Figs. 2 and 3a). This occurs in our scenarios as repeated drops in adherence over time may reduce the efficacy of a lockdown to values lower than a simple curfew after a few weeks, because of the small difference between the estimated efficacies of the second lockdown (before relaxation) and curfew conditions. Since moderate lockdowns would not be able to considerably reduce viral circulation, they would entail a larger impact on the hospital system (median hospitalizations in the period w12–w26 around 38,000–50,000 compared to 10,000–23,000 for strict lockdowns, Fig. 3c) for a longer time (median 6–10 weeks with hospitalization incidence above the peak of the second wave compared to at most 2 weeks for a strict lockdown of any duration, Fig. 3d). This impact would be more substantial if adherence waned, leaving the hospital system under high pressure for twice the amount of time (median 12 weeks above the peak of the second wave assuming continuous adherence loss, compared to 6 weeks for full adherence, corresponding to 80% of the time period under study).

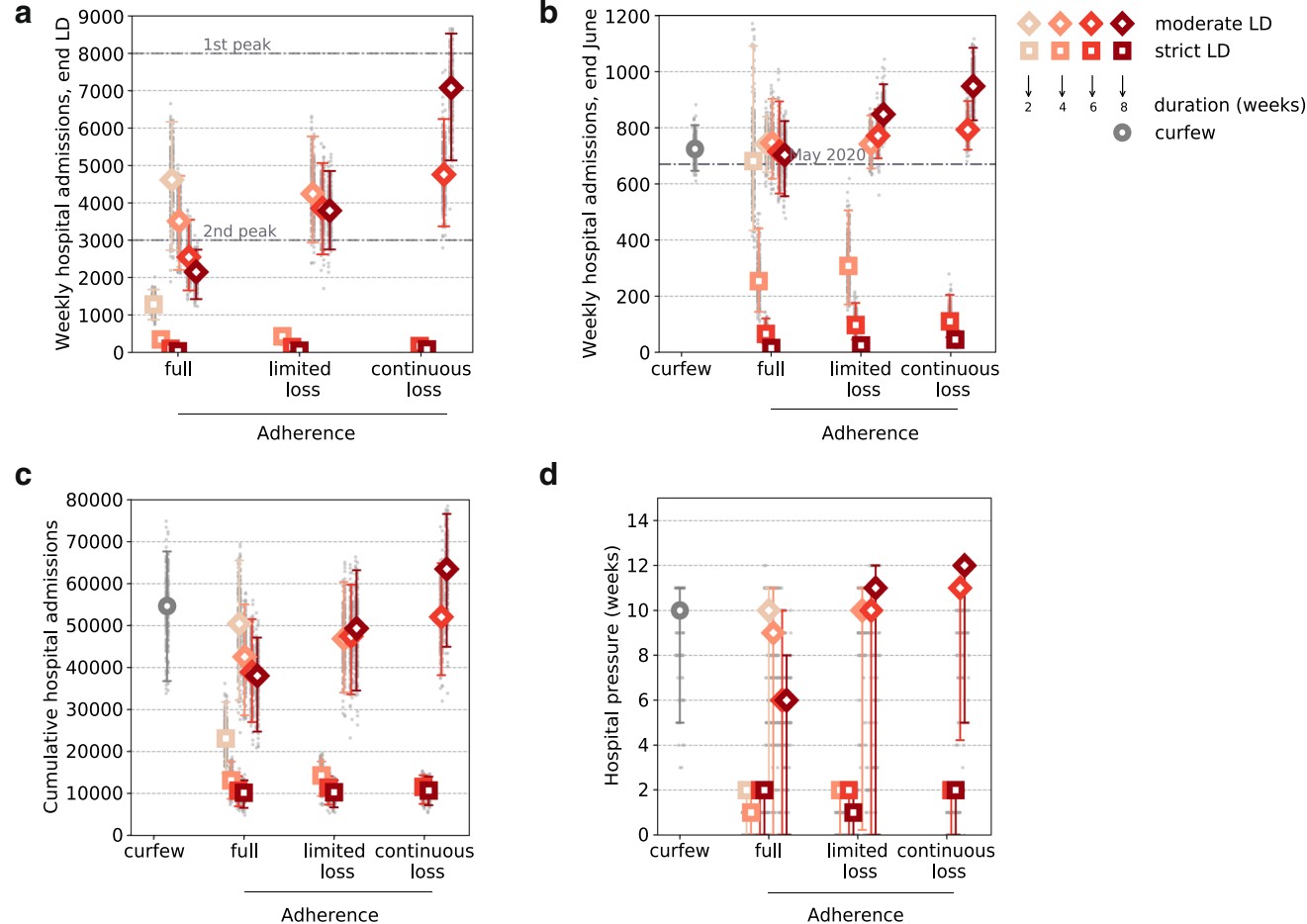

**Fig. 3 Impact of loss of adherence on intervention efficacy, for varying stringency and duration of interventions.** Weekly hospital admissions at the end of the lockdown (**a**), weekly hospital admissions at the end of June (w26) (**b**), cumulative hospital admissions computed over the time period w12–w26 (**c**), hospital pressure, defined as the number of weeks in which hospital admissions remain above the peak level achieved during the second wave, in the period w12–w26 (**d**) as functions of the adherence level—full adherence over time, limited loss of adherence, continuous loss of adherence over time. The point with the curfew (gray circle) represents the estimate under the curfew scenario with no additional intervention, and is shown for comparison. Results refer to a vaccination rhythm accelerated to 300,000 first doses/day since April. Symbol types refer to the stringency of intervention (squares representing a strict lockdown scenario, diamonds representing a moderate lockdown scenario). Color shades of the symbol contour refer to the duration (weeks) of the lockdown intervention (from the lightest shade corresponding to 2 weeks, to the darkest one corresponding to 8 weeks); the abbreviation LD in the legend stands for lockdown. Plots show median values; error bars represent 95% probability ranges, obtained from n = 250 independent stochastic runs (gray points). Horizontal dashed lines in panel (**a**) refer to the peak of the first and second wave in the region; horizontal dashed line in panel (**b**) refers to the level of mid-May 2020 at the exit of the first lockdown.

Despite different trajectories, our model anticipates that moderate lockdowns could reach at the end of June the hospitalization levels measured in May 2020 (670 weekly admissions) if adherence was maintained over time, similarly to a short strict lockdown, and with no advantage compared to curfew measures at this stage (Fig. 3b). Adherence loss would lead to higher hospital admission levels.

Removing contacts at schools, we found that the two weeks of school closure for spring holidays (w15–16) would have a marked impact on the efficacy of moderate lockdowns, otherwise with schools open. They would allow flattening the curve and avoiding even longer plateaus in critical conditions before the accrued effect of immunization would decrease the epidemic (Supplementary Fig. 8).

**Optimizing interventions' sustainability by minimizing policy-induced distress.** Another critical dimension associated with the nature of interventions—besides their stringency, duration, and adherence—is their sustainability over time, which is a combination of intensity of restrictions and how long they last. To account for this aspect, we introduced a distress index, integrating the intensity, duration, and adherence level in each scenario, and providing a quantitative measure of the policy-induced distress perceived on average by an individual. The higher the distress index and the lower is the sustainability of the measure.

Moderate lockdowns of less than 6 weeks are all characterized by low levels of distress (<4), similar to those of a curfew and of a 2-week strict lockdown (Fig. 4; estimated values of the distress index are reported in the Supplementary Table 2). In this range of distress values, a net advantage was observed for the short strict lockdown that substantially reduced the total number of hospitalizations (23,000 vs. an average of 47,000) and hospital pressure (2 weeks vs. more than 8 weeks). High values of the distress index (>7) were associated exclusively to strict and long lockdowns (of 6 weeks, with full adherence, or longer, also with adherence waning over time), which correspond to the most effective measures in suppressing viral circulation and reducing the healthcare impact, but also the least sustainable.

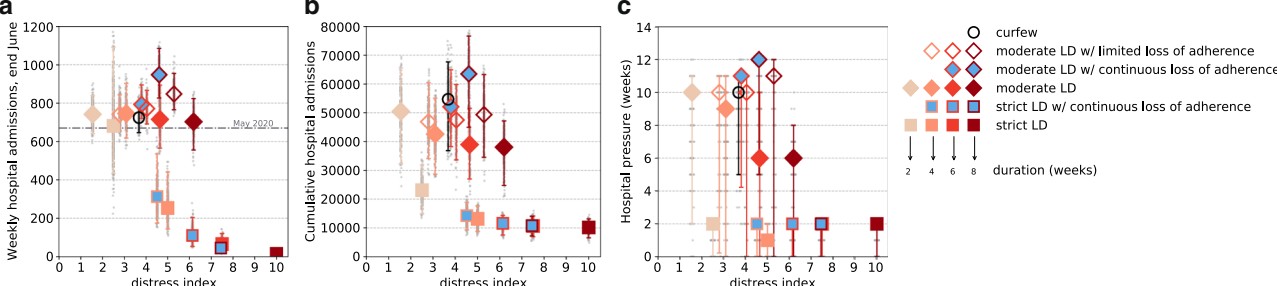

**Fig. 4 Intervention efficacy vs. associated policy-induced distress.** Weekly hospital admissions at the end of June (w26) (**a**), cumulative hospital admissions (computed in the time period w12–w26) (**b**), hospital pressure, defined as the number of weeks in which hospital admissions remain above the peak level achieved during the second wave, in the period w12–w26 (**c**) as functions of the distress index. Results refer to the accelerated vaccination pace of 300,000 first doses/day since April. Symbol types refer to the stringency of intervention (squares representing a strict lockdown scenario, diamonds representing a moderate lockdown scenario, void circle representing the projection under the curfew scenario). Color shades of the symbol contour refer to the duration (weeks) of the lockdown intervention (from the lightest shade corresponding to 2 weeks, to the darkest one corresponding to 8 weeks); the abbreviation LD in the legend stands for lockdown. Adherence to moderate and strict lockdowns is coded with the fill color (filled symbols with the color of the scenario correspond to scenarios with full adherence, void symbols represent scenarios with limited loss of adherence, blue filled-in symbols correspond to scenarios with continuous loss of adherence). Plots show median values; error bars represent 95% probability ranges obtained from $n = 250$ independent stochastic runs (gray points). Horizontal dashed line in panel (**a**) refers to the level of mid-May 2020 at the exit of the first lockdown.

There exists, however, quite a diversified range of intervention options that, for moderate distress (index between 4 and 7), would achieve better control of the epidemic than moderate lockdowns. One-month strict lockdowns would largely outperform moderate interventions in terms of health metrics while inducing similar distress, as the latter must be maintained for a longer duration. Six- or eight-week moderate lockdowns would lead to about three times as many patients hospitalized and about three to six times the hospitalization incidence at the end of June compared to interventions that exhibit a similar distress level as a strict lockdown of 1 month. If moderate lockdown restrictions were less respected over time, epidemiological and healthcare indicators would considerably worsen, for relatively small gains in lowering the policy-induced distress.

**Vaccination and seasonality while managing reopening plans.** According to weather data for Île-de-France, an average increase of 23 kJ/m$^2$ in UV radiation and of 11 °C in temperature were registered from March to June in the last three years. Based on the estimated relation between climate factors and the reproductive number[40,41], this increase would correspond to a 7.7% and 7.3% reduction in viral transmission, respectively. In the following, we explore values up to 30% reductions of the transmissibility starting from May, to also account for reductions resulting from changes of behaviors associated with the upcoming summer (e.g., more time spent outdoor, increased ventilations of indoor environments, etc.).

Further acceleration of vaccination pace coupled with the potential effect of seasonality may act in synergy to (i) counteract the deterioration of the epidemic due to the waning of adherence over time, or (ii)—if stronger—bring down the epidemic faster than what is expected from moderate interventions. Figure 5 shows the interplay of these factors assuming that seasonality acts on reducing viral transmission starting May. Keeping the planned vaccination rhythm at 300,000 first injections per day since April, a 5–10% reduction of transmission induced by seasonality would be necessary to absorb the potential loss of adherence against moderate interventions by the end of June (label (1)). Without counting on seasonal effects, vaccination rollout should increase by 33% (i.e., from 300,000 to 400,000 first doses per day). Larger seasonality (>20%) or accelerations in vaccination rollouts (up to 800,000 first doses per day) would be able to compensate for the larger cumulated number of patients requiring hospitalization

due to adherence loss (label (2)). Reaching by the start of the summer the weekly admissions achieved by an imperfectly adhered 1-month strict lockdown would require substantial seasonality coupled with large increases in vaccination rhythms (contour line at 300 in the top row of Fig. 5).

In all situations, a certain degree of social distancing is required to accompany the gradual lifting of lockdown to avoid slowdowns or rebounds (bottom row of Fig. 5). Even the summer conditions estimated in mid-July 2020, but considering schools in session, may lead to an epidemic resurgence if incidence is high, despite the growth in population immunity and summer seasonality. Results show that a progressive transition in phasing out restrictions is essential, and they further support the importance to lower the incidence level to better manage potential rebounds while reopening.

## Discussion
Managing sustained viral circulation after long periods of social distancing measures of varying intensity faces the challenge to reduce the strain on the healthcare system and to limit long-lasting or stringent interventions affecting the quality of life of the population. Moreover, with accelerating vaccination campaigns and the prospects of reopening the society, adherence waning may represent a threat to phasing out restrictions. Using Île-de-France as a case study, we compared the efficacy of different measures against their sustainability and potential for case resurgence due to imperfect adherence of the population. Given the high incidence levels reached by the epidemic in the region by mid-March 2021, exceeding the peak of the second wave[10], only high intensity interventions would have been able to rapidly curb viral circulation, allowing the region to considerably reduce the burden of hospitalization after only 2 weeks and despite loss of adherence. Once incidence substantially declined, the management of the epidemic could largely benefit from test-trace-isolate strategies[8,43] and the large-scale availability of self-test kits for iterative screening[45], while immunization due to vaccination builds up in the population. Hospitals could more rapidly restore routine care beyond COVID-19. Moreover, rapidly reaching low incidence levels would also lower the potential for SARS-CoV-2 evolution conferring fitness advantages, and allow a better control of the possible emergence or importation of variants of concern[46].

Moderate interventions as the strategy adopted in November–December 2020 to curb the second wave constitute

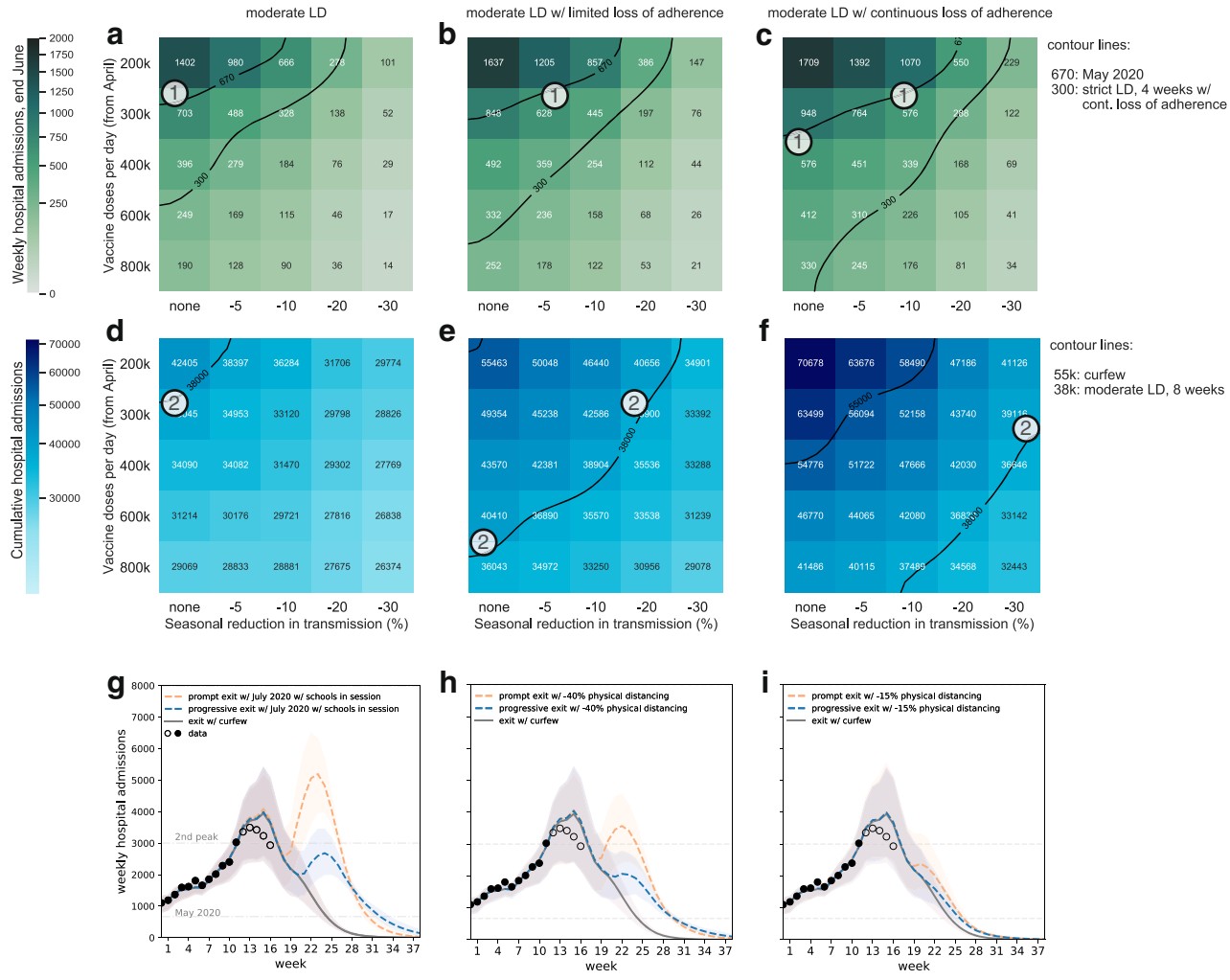

**Fig. 5 Impact of vaccination, seasonality, and reopening plans. a–f** Heatmaps show median values of weekly hospital admissions at the end of June (w26) (**a–c**) and cumulative hospital admissions in the time period w12–w26 (**d–f**), as functions of vaccination rhythm (*y*-axis) and seasonal reduction in transmission (*x*-axis) for moderate lockdowns of 8 weeks with full adherence (**a, d**), limited loss of adherence (**b, e**), continuous loss of adherence (**c, f**). The abbreviation LD stands for lockdown, contour lines indicate reference values of specific scenarios defined in the legends. Circled numbers refer to a subset of conditions of interventions (heatmaps from left to right), vaccination and seasonality (variables in the *y*- and *x*-axis in each heatmap) achieving the same outcome, identified by the contour lines (see legend) and discussed in the main text. **g–i** Plots show projections of the weekly hospital admissions under different hypotheses for the reopening conditions, assumed right after lifting the moderate lockdown (orange curves), or through a progressive transition (blue curves): conditions experienced in mid-July 2020, but with schools in session (**g**); curfew scenarios with 40% fewer individuals respecting physical distancing (**h**); curfew scenarios with 15% fewer individuals respecting physical distancing (**i**). Curves and shaded areas correspond to median trajectories and 95% probability ranges, obtained from *n* = 250 independent stochastic runs. In all plots, gray continuous line refers to a scenario in which curfew conditions as in week 11 are restored after the moderate lockdown. Dots refer to data; filled dots correspond to the data used to fit the model and to provide the trajectory for the curfew scenario; void dots correspond to data outside the inference time window showing the peak of the third wave. Scenarios assume a 10% reduction in transmissibility due to seasonality (except for the mid-July 2020 conditions that already embed seasonal aspects) and a vaccination rhythm of 300,000 first doses per day starting April. Plots showing projections for the reopening conditions, assumed right after lifting the strict lockdown are shown in Supplementary Fig. 10.

suboptimal options for the management of the epidemic till summer. Their efficacy remains limited because of the Alpha variant's higher transmissibility[1,2,29] and still low immunization levels (15.7% population vaccinated with a first dose in the region by April 20). Our results show that these measures should be maintained for much longer to reach incidence values similar to the result of a short and strict circuit-breaker[44], at the expense of a large number of severe cases requiring hospital care, a continuously high pressure on the hospital system, and high levels of distress cumulated over time. The strengthened measures in place during the spring 2021, based on closure of non-essential businesses, ban on gatherings and recommendations to telework, are

similar in intensity to the moderate scenarios considered in this study, as also confirmed by the similarity of the stringency index[47] (Supplementary Fig. 4). These were however accompanied by the advanced closure of schools just before the 2 weeks regular school holidays in April that provided an extra break on the epidemic evolution showing in the observed trend.

Despite different trajectories, epidemic conditions by the time summer starts are predicted to be similar across intervention scenarios (with the exception of high intensity interventions lasting 1 month or longer, largely suppressing the epidemic) and close to the curfew scenario in absence of additional restrictions. Differences in how the epidemic is managed throughout spring

are absorbed over time thanks to vaccination. However, large disparities remain for cumulative epidemiological and public health indicators, depending on whether early suppression or mitigation were achieved by the interventions. This would have an impact not only throughout the third wave (by increasing the overall number of hospitalizations, patients requiring critical care, and deaths), but also on the medium-to-long term due to the rising number of individuals who are likely to suffer from long-term health consequences following a COVID-19 infection (long COVID)[48,49]. Early estimates indicate that about 10% of individuals testing positive for COVID-19 exhibit symptoms after 4 months[48], and about 3/4 of hospitalized patients report at least one symptom after 6 months[49]—mainly fatigue, muscle weakness, sleep difficulties, anxiety, depression. Choosing a 2-week strict lockdown against an 8-week moderate lockdown would correspond to estimated 30,000 avoided long COVID cases among detected infections from mid-March to the end of June.

The choice of interventions inevitably also impacts the quality of life of the population due to imposed restrictions, leading to possible spontaneous relaxation. The shorter the measure's application, the less likely it is to observe adherence waning over time. Interventions of high intensity but short duration may therefore constitute an optimal approach to reduce both epidemic and healthcare burdens, while minimizing possible loss of adherence as well as policy-induced distress. Evidence from OECD countries after one year of COVID-19 pandemic show that swift lockdown measures were overall less restrictive of civil liberties, thanks to achieved control, compared to recurrent mitigation policies severely impacting individual freedom[50]. Indeed, moderate or mild (curfew) interventions cumulate distress over time, as they need to be implemented for much longer to achieve the reduction of health indicators, with the potential risk of losing population adherence. This would considerably worsen both incidence and cumulative indicators, slowing down or stopping the decrease in incidence obtained with restrictions, thus remaining on a long plateau at sustained viral circulation, as occurred after the second wave. If relaxation against measures is left uncontrolled, epidemic rebounds can also be expected. At the same time, loss of adherence would correspond to a limited gain in personal freedom, when averaged overall individuals (−26% in distress index by continuously losing adherence in moderate lockdowns lasting 8 weeks), compared to interventions of lower stringency (−60% in distress index from an 8-week moderate lockdown to a 2-week strict lockdown).

In our study, loss of adherence occurs over time and is informed from observed increases in mobility during the second lockdown and corresponding estimated impact on the epidemic, during unchanged restrictions and recommendations. We did not consider initial adherence to restrictions different from what was measured in the first and second lockdown. While lower initial adherence may be expected as stringent social distancing measures are being applied for the third time, this may also depend on the acceptability of new measures, clarity of restriction and recommendations. For example, a recent survey showed that about 70% of individuals approved the strengthened measures recently applied in France, however almost half of them planned to disobey the rules[51]. Also, adherence loss should not be confused with population response to restrictions induced by socio-economic conditions and life circumstances[6,12,13,52]. Prior work showed that this response—despite numerical differences depending on the stringency of measures (first lockdown, second lockdown, localized curfew at 8 pm, nationwide curfew at 6 pm)—is associated to the composition of the population, with blue-collar jobs and household crowding emerging as determinants of higher mobility during restrictions in France[13].

We introduced an index integrating mobility reduction and duration of restrictions to provide a quantitative measure of policy-induced distress along the spectrum of varying stringency. This is meant to integrate the impact of restrictions infringing on individual freedom, as well as psychosocial effects of prolonged measures, linked for example to isolation, uncertainty, loss of purpose, and lack of social contacts[53,54]. While both distress index and adherence were informed from data, we did not consider an explicit relation between distress and adherence loss, potentially leading to feedback mechanisms reinforcing relaxation for increasingly long durations. Related to "pandemic fatigue", a concept often introduced as the presumed cause to limited adherence, this relation remains highly debated[6]. Some behavioral scientists warned against an ill-defined concept used to justify avoiding strict and/or early interventions[6,55]. Different features and origins of fatigue are likely at play—including for example life constraints independent of motivation, as discussed above—that would require a range of definitions, data, and frameworks for analysis. A study on data from 14 countries showed that adherence to physical distancing evolved following a U-shape between March and December 2020[5]. However, in France this drop would correspond to the summer period, between the first two waves, during which restrictions were lifted and only recommendations on the use of personal preventive measures were in place. As such, it does not relate to the adherence loss throughout interventions considered here. Different indicators obtained from surveys show that fear of contracting COVID-19 decreased over time after the second wave in France, while anxiety continued to increase in the population. We found a positive association between fear and social distancing (expressed by the percentage of individuals avoiding crowded places), confirming the role that risk perception has in shaping health-related behaviors[56]. However, we did not find a significant association between increasing anxiety, concurrent with lasting restrictions, and decreasing social distancing (Supplementary Fig. 5). So far there exists little evidence on the mechanisms of action of behavioral interventions that could improve our understanding and be leveraged to boost policy observance.

Available evidence indicates that interventions implemented in 2020 largely reduced the incidence of COVID-19[9,11,57–60], in the absence of effective treatments and prior to vaccination. Substantial differences were observed between analyses aiming to assess the efficacy of single social distancing measures (e.g., closure of schools, businesses, all but essential services, ban on mass gatherings and public events, restrictions on movements and stay-at-home orders). Our study did not focus on isolated measures, but considered the estimated efficiency of policy packages that were deployed during the first and second wave in France, along with observed policy compliance and wane in time. A lockdown as strict as the first one is unlikely to reach nowadays the efficiency observed last year, and for this reason we considered reduced adherence, which we show would marginally affect the results. The two lockdowns implemented in 2020 did not differ exclusively for the closure or opening of schools, but also for the mobility levels and presence at workplace estimated from data. Behaviors related to mobility, presence at work and school are not independent and we currently lack enough data to parameterize their relationship. In addition, alternative versions of interventions allowing time outdoor where risk of transmission is reduced[61]—such as recommendations in place during the third wave—may reshape mobility, contacts and associated risk in ways different than previously observed, preventing their assessment on the basis of historical data. Open questions remain on the combination and sequence of restrictions to be progressively lifted after the lockdown, as specific measures are too detailed for

mathematical models to quantify (e.g., reopening of restaurants). Strategic prioritization will likely depend on countries' interests.

Vaccination is key to exit the health crisis; however, our numerical evidence shows that epidemic management still needs to rely on social distancing to curb viral transmission, confirming prior work[62–64]. Increasing vaccination rollout coupled with 5-30% reduction in transmission due to seasonal effects would be able to compensate for the slowdown or rebound effects of adherence waning or fast reopening. Multiple studies have investigated the relationship between SARS-CoV-2 transmission and weather. Results suggest that warm and humid conditions, and high UV radiation levels, are less favorable to disease spread[39]. Based on previous estimates[40,41], we derived that the average increase in UV radiation and temperature reported in Île-de-France from March to June corresponds to ~10% reduction in transmission. Additional mitigating effects are expected due to seasonal behavior, with individuals spending more time outdoor than indoor, and aerating indoor settings more compared to winter time. But misconceptions on seasonality may generate excessive trust in the public altering their risk perception, and in authorities affecting their decision-making[65]. Despite a building literature on the topic, there remain aspects that are difficult to measure and include a strong behavioral component. A large second wave started last year in the United States during summer because of early reopening, and cases started to rise in France from mid-July 2020, paving the way to the second wave in the fall. Lifting restrictions with the conditions experienced in mid-July 2020 is expected to lead to an epidemic rebound if incidence is high. We did not consider here the situation at the end of the first lockdown in spring 2020 because it was characterized by the maintenance of cautious behaviors, and additional levers existed that continued curbing transmission after lockdown was lifted (e.g., the increase in mask use, from 45% in mid-May 2020 to >70% at the end of the summer[17], also due to mask mandates). Managing the epidemic while gradually releasing non-pharmaceutical interventions through the summer should mainly rely on the speed of vaccination rollout.

Our study has a set of limitations. It is applied to a region only, as indicators for France hide a variable situation at the local level, limiting the accuracy of modeling approaches extended to the whole country. Geographical heterogeneity depends on the evolving epidemic situation, population immunity due to natural infection, and variant frequency, so that results are not directly generalizable to other regions. We did not consider waning of immunity[66] or reinfections over the time frames modeled. We assumed the transmissibility advantage of the Alpha variant from early estimates in France[29], in agreement with other studies[1,2]; however, this may be altered over time by social distancing and competition with other strains. Assuming a smaller transmissibility advantage for the variant would lead to lower incidence projections; however, it would not be able to capture the evolution in time of the Alpha variant's frequency in the region (Supplementary Figs. 3, 11). We did not consider the interaction with other variants, such as the Beta variant or the Gamma variant, that are already present in the country and show so far limited diffusion. If these variants can at least partially escape natural or vaccine-induced immunity[67], they may pose a challenge for the management of the epidemic as population immunity increases. Our approach is not suited to account for contacts in low-risk and high-risk conditions, e.g., in closed ill aerated settings vs. open settings, but seasonal reductions effectively account for these aspects. Modeled vaccination rhythms according to authorities' plans were slightly faster than observed. By May 4, 23.6% of the population was vaccinated with a first dose in the model, compared to 20.3% according to data; however, this is not expected to affect our findings. We did not

consider slowdowns that were recently observed after the temporary stop of AstraZeneca vaccine administration, undermining demand relatively to other vaccines. We considered 50% coverage in the adult population, following the declared intentions to get vaccinated of this age class in France[32], but we did not consider changes in this expected coverage due to a possible reduction in perceived risk in relation to the successful reduction of epidemic incidence[68] or the application of measures targeting the non-vaccinated population thus incentivizing uptake. Our findings and prior work show that relaxing social distancing with limited immunization may result in epidemic rebounds[62–64]. We did not consider the economic impact of social distancing measures, as our study focused on the epidemiological, healthcare, and behavioral components. There is increasing evidence, however, that economic growth, public health, and civil liberties do not need to be in opposition in the management of the COVID-19 crisis, with countries aiming for elimination faring largely better than countries adopting mitigation strategies[50]. Also, we did not consider health impacts beyond COVID-19 that can result from a high pressure on the hospital system. Psychosocial impact was instead introduced through a simplified empirically-driven indicator based on restricted mobility, the core of many social distancing measures. However, this indicator is an average, therefore it hides the effects on vulnerable populations who may experience disproportionately higher distress[6,13,52]. Also, being informed by mobility only, it aims at providing a measure of infringement of personal freedoms, but without explicitly capturing other elements associated with the quality of life[54]. However, the increasing trend in anxiety observed following the second wave and throughout a prolonged application of curfew measures supports the idea of a progressive buildup of distress concurrent with lasting restrictions.

Control of the epidemic in a partially immunized population depends, in non-linear ways, on the interplay between the characteristics of the circulating variants, the stringency of social distancing measures, vaccination rollout plans, and population adherence to measures and vaccination. Mathematical models help to unravel the complexity of these interactions, accounting for the uncertainties characterizing some of these aspects, and to quantitatively inform on the optimal solutions for epidemic control. Our study shows that favoring milder interventions over more stringent approaches limited in time on the basis of perceived acceptability could be detrimental in the long term, especially with waning adherence.

## Data availability

The mobility data supporting the findings of this study were available to authors from the Orange Business Service Flux Vision within the framework of the research project ANR EVALCOVID-19 (ANR-20-COVI-0007). Restrictions apply to the availability of these data, which were used under license, and so are not publicly available. Access to the data can be requested from Orange Business Service Flux Vision on a contractual basis. All other indicators used in the study are publicly available online at the links provided in the references. Hospitalization data were obtained from the SIVIC dataset[7]. Presence at workplaces was obtained from Google Mobility Reports[15] specific to Île-de-France region. Indicators of social distancing ("Avoiding crowded public places")[69] and risk perception ("% people who say they are 'very' or 'somewhat' scared that they will contract COVID-19")[70] were obtained from YouGov.uk. Data on mental health were obtained from Santé publique France[17], in the section "Santé mental - Prévalences et évolutions de l'anxiété". Source data for the main figures in the manuscript can be accessed as Supplementary Data 1-5.

## Code availability

Analyses were carried out in Python 3.8.5. Code for the transmission model is available on GitHub[71].

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

## Acknowledgements

This study is partially funded by: ANR projects DATAREDUX (ANR-19-CE46-0008-03), EVALCOVID-19 (ANR-20-COVI-0007), and SPHINX (ANR-17-CE36-0008-05); EU H2020 grants MOOD (H2020-874850) and RECOVER (H2020-101003589).

## Author contributions

V.C. conceived and designed the analysis. L.D.D. and C.E.S. performed the analysis. V.C. wrote the manuscript. L.D.D., C.E.S., P.Y.B., C.P., P.C., J.P., S.C., F.B., H.N., D.L.B. and V.C. critically revised the manuscript and approved its final version.

## Competing interests

The authors declare no competing interests.
