## [Peer Review File · Communications Medicine]

Reviewers' comments:

Reviewer #1 (Remarks to the Author):

The authors provide a modeling study addressing an important question of balancing the length and stringency of control measures with their efficacy. The authors consider the third wave in France when the B.1.1.7 variant was circulating and vaccines were being distributed. Overall the problem is important and the model appears to incorporate important features of COVID-19. However there are some gaps in the model that are not well-explained, making it challenging to determine if the results are reliable.

1. It is not entirely clear what parameters are being estimated and what are being assumed known.
2. In some cases it appears that the transmission parameter, β , is being estimated, but then it is stated that there is an estimated effective reproductive number (lines 542-551). Since these two are really related, it is not clear how the estimated effective R_t are being derived and/or used in the model. I am having a hard time understanding that from what is written-it seems that R_t is being used as an input in the model, for instance to capture waning compliance with interventions. If that is the case, these estimates should be explained. If R_t is an input in the model, then estimating β would not make a lot of sense because these two quantities are so closely related.
3. Line 569-570. Related, it is stated that R_t is increased 10.9% to capture loss of adherence. This does not appear justified by data.
4. Distress index. Is this not time varying during the period of interventions? I am wondering if the distress index is the same on week 1 of lockdown compared to week 5. Is there any sense that this would be time-varying? Also, it would be helpful to see if this metric corresponds with other validated measures of anxiety or disruption-i.e. is there a better way to justify this metric?
5. Line 600-601. It is not clear why UV index values and temp values stated correspond to the decreases in transmission that are stated. Separately, is this implemented as a decrease in R_t which is input into the model? Again, the use of R_t is unclear.
6. Lines 64-78. The paragraph starts by describing how the authors will fit the model to the first 11 weeks of data and then project different scenarios. After that, the paragraph seems to veer into discussing historical trends. This is a bit confusing and unclear presentation. Also are weeks 11 and 12 in 2021 and data from lockdowns used to model interventions starting in week 12 of 2021? The presentation could be a bit more clear.
7. I assume the model was initialized with a population to match the area being studied, but I cannot see this specified anywhere (apologies if I missed it) and what data was used to initialize the population.

Minor

1. The paper would benefit from careful editing-there are some grammatical errors and statements that do not make much sense.
2. Line 526. I think $H_{pred}(t)$ be $H_{pred}(t|\Theta)$, else likelihood does not make much sense.
3. Helpful to explain how week 11 and 12 correspond to actual dates.

Reviewer #2 (Remarks to the Author):

In this paper Di Domenico and colleagues study a range of potential scenarios for interventions to control the "third wave" of COVID-19 in Paris, France, which I believe roughly corresponds to the

time period of spring and summer 2021. Mostly the scenarios include different combinations of generalized social distancing (which correspond to different degrees of maximal reductions in contacts, different amounts of loss of adherence to these policies over time, and different total durations of these “lockdowns”) and different rates of vaccine administration. They compare the number of expected hospitalizations due to COVID-19 in each scenario. In some analyses they compare this benefit of interventions with a measure of their cost to society, using a “distress” metric they invent that incorporates both the strength and duration of contact reductions. The scenarios they consider are motivated by the degree of contact reduction seen during lockdowns imposed for the “first” (spring 2020) and “second” (fall 2020) waves of COVID-19 in France.

I assume these authors are involved in very applied modeling efforts to regularly advise the French government on policy options using very data-driven/region-specific models, and that these analyses were originally conducted for that purpose. This paper seems to be the result of rebranding those analyses to say something more general about the cost/benefit trade-off between stronger but shorter interventions compared to those that are less stringent but last longer.

I think the strengths of this paper are

- * that the authors use a well-established and previously published model of COVID-19 transmission (their own) which is carefully parameterized based on data from the literature
- * the authors use a well-defined model fitting procedure which only estimates a single parameter per time period
- * the modeled impact of interventions on transmission rates is informed by real data on past interventions
- * when modeling the implementation of social distancing, the authors take into account waning of adherence to these interventions over time, again based on real data from early in the COVID-19 pandemic
- * the study attempts to quantify the social cost of interventions
- * the consideration of scenarios is very thorough: the authors considers many possible combinations of different sorts of interventions of different strengths, instead of just varying one at a time

As the authors claim in their title and abstract, their results do indeed support the idea that stricter short-term lockdowns are more effective than long-term partial restrictions. Compared to other work that has come to the same conclusion, this paper has the benefit of using real data on the impact of policies on mobility and how they wane over time to inform their conclusions. Though as the authors themselves admit, it is difficult to know how general these findings are, since many aspects of the model are very specific to the geography and time period they consider. I do not have any concerns about the technical validity or the statistical methods used in the paper.

Overall I think the analyses are solid and the results interesting. The main weaknesses of the paper were in the organization and presentation of the results. I found the paper quite hard to follow. Initially, I was quite confused about what results came from modeling vs from data, and which precise period was used for model calibration vs model projections. For the first half of the paper, I think things could be clarified a lot by subdividing it further into sections and being careful not to jump around too much between topics.

I also have a few minor comments about confusing aspects of the figures:

- * Abstract/Introduction: Early on and throughout the paper the authors should be more clear about the time periods being considered. When does model fitting start and end? (2020? 2021?). When

does forecasting begin and end? I think it is especially confusing because there seems to be overlap in the time periods you present data for versus those you are projecting. For example, Figure 1 goes beyond week 12 with data for 2021 but then I think week 12 is when you start projecting? If possible, I would suggest using real dates, with day/month/year, not weeks, which are quite unintuitive and it is often unclear which year you are referring to. Not everyone will be familiar with when the “first wave”, “second wave” and “third wave” occurred in Paris, since waves were out of sync around the world.

* There were a few words I had not heard before: performant, perduring

* Figure 1 - it's confusing that the x axis on the bottom row does not line up with the top two rows

* Figure 2 - it's really hard to see differences in colors. The legend suggests there are 6 different scenarios but in most plots I only see only 4 or 5 lines. And what is the curfew scenario? (I don't think this was ever explained)

* Figure 3 - Is the shading of the plot markers really necessary? It is hard to see, and the x axis location gives the adherence pattern. I found these plots very hard to read with so much going on. It was hard to interpret the meaning of the symbol shape/color.

* Figure 5 - The plots in the lowest row seem totally separate from those in the upper 2 rows ... maybe they should be a separate figure? Also the legends for the bottom row are confusing. It's really hard to tell differences between 2 line colors and what they represent. The labels on the figure didn't say this, and I had to dig through caption to figure it out

POINT-BY-POINT RESPONSE TO REVIEWERS

Referee #1:

The authors provide a modeling study addressing an important question of balancing the length and stringency of control measures with their efficacy. The authors consider the third wave in France when the B.1.1.7 variant was circulating and vaccines were being distributed. Overall the problem is important and the model appears to incorporate important features of COVID-19. However there are some gaps in the model that are not well-explained, making it challenging to determine if the results are reliable.

We thank the reviewer for the positive assessment of our study and for the opportunity to explain certain aspects of the model that were not sufficiently clear in the original version. Point-by-point replies to the remarks raised by the reviewer are reported below.

1. It is not entirely clear what parameters are being estimated and what are being assumed known.

The inference framework is described in the Methods section; overall, we estimate only the transmission rate through a maximum likelihood approach. More in detail, in the early 2020 phase prior to any intervention, we estimate the transmission rate β (pre-lockdown transmission rate) and the starting date of the simulation. Then, for each pandemic phase (e.g. first lockdown, second lockdown, etc.) we estimate the scaling factor of the pre-lockdown transmission rate. From this estimate, we compute the effective reproductive number through the next generation matrix approach (see reply below). All other parameters used to define the compartmental model are informed from the literature and reported in the Methods section and in the Supplementary Information.

2. In some cases it appears that the transmission parameter, β , is being estimated, but then it is stated that there is an estimated effective reproductive number (lines 542-551). Since these two are really related, it is not clear how the estimated effective R_t are being derived and/or used in the model. I am having a hard time understanding that from what is written—it seems that R_t is being used as an input in the model, for instance to capture waning compliance with interventions. If that is the case, these estimates should be explained. If R_t is an input in the model, then estimating β would not make a lot of sense because these two quantities are so closely related.

From the estimated transmission parameter β , we compute the effective reproductive number through the next generation matrix approach (Diekmann et al., 2010). This is now included in the Methods section. The next generation matrix approach takes as input the estimated transmission rate, the values of the epidemiological parameters, the susceptible population and the social contact matrix specific to the pandemic phase. In the text, we refer to the reproductive number (instead of the transmission rate) as it allows a straightforward interpretation.

From the fit of the model to the entire pandemic history, from early 2020 to the start of the third wave in France (week 11, mid-March 2021), we estimate the reductions in the transmissibility (and therefore R) induced by the first lockdown and by the second lockdown. In addition, during the second lockdown, we also estimate the impact of relaxation against restrictions on the value transmissibility (and therefore R). We used all these estimates to inform the scenarios for the third wave, starting from week 12 2021, and considering hypothetical lockdowns of different stringency (based on the first or second lockdown), duration, and waning adherence (based on estimates from the second lockdown).

3. Line 569-570. Related, it is stated that R_t is increased 10.9% to capture loss of adherence. This does not appear justified by data.

The relative increase of 10.9% in the reproductive number used to model waning adherence in the scenarios under study is informed from estimates on the second lockdown. During the second wave, trends in mobility and presence at work showed a rapid increase towards the end of the second lockdown compared to its first weeks of application (see **Figure 1**), suggesting a loss of adherence over time. By fitting the model to the second lockdown data, we found that this corresponded to a relative increase of 10.9% in the reproductive number estimated in the fourth week, with respect to the reproductive number estimated in the first three weeks of the lockdown.

4. Distress index. Is this not time varying during the period of interventions? I am wondering if the distress index is the same on week 1 of lockdown compared to week 5. Is there any sense that this would be time-varying? Also, it would be helpful to see if this metric corresponds with other validated measures of anxiety or disruption-i.e. is there a better way to justify this metric?

The distress index is not time-varying, as it is meant to be a cumulative measure of the distress induced by a lockdown period. As such, it takes into account the total duration of the lockdown, the intensity of the intervention and the level of adherence. We chose a cumulative measure in order to allow comparisons between lockdown strategies that may differ in duration (besides in stringency and acceptance). In other words, we wanted to introduce a measure to be able to respond to the question “is it better to implement a shorter strict lockdown or a longer moderate lockdown?” in terms of individual perception of the restrictions (more stringent but shorter vs. less restrictive but longer).

In the paper, we also consider mental health indicators, measured by the prevalence of anxiety due to the COVID-19 pandemic (data shown in the third row of **Figure 1**). These estimates are the result of self-reported data over time. They cannot be used as a replacement of the distress index as they provide an instantaneous indication of the impact induced by the restrictions, instead of a cumulative one as the distress index. However, the increasing trend in anxiety observed following the second wave and throughout a prolonged application of curfew measures supports the idea of a progressive buildup of distress concurrent to lasting restrictions. This is now explicitly discussed in the text.

5. Line 600-601. It is not clear why UV index values and temp values stated correspond to the decreases in transmission that are stated. Separately, is this implemented as a decrease in R_t which is input into the model? Again, the use of R_t is unclear.

The decrease in transmission rate is computed based on the estimated dependence of the reproductive number on UV radiation (Metelmann et al., 2021) and temperature (Wang et al., 2021). Metelmann et al. found that, starting from a reproductive number around 1.5, an increase of 1 kJ/m² in UV radiation corresponded to a decrease of 0.005 in the reproductive number. Wang et al. found that, with a reproductive number around 3, an increase in temperature by 1°C is associated with a reduction in reproductive number by 0.02. We used these relations to estimate a relative reduction in the reproductive number due to seasonality based on the increase in UV radiation and temperature observed in Île-de-France from March to June in the last three years, yielding an estimate of 7-8% average variation. These estimates are not meant to be used as model input, given their limitations, but rather as indicators of a plausible range of values to be explored. When studying the impact of seasonality, we chose 5% as a lower bound and explored higher values up to 30% reduction to test the impact of a stronger effect due to seasonality (also induced, for example, by changes in behaviors such as spending more time outdoors).

6. Lines 64-78. The paragraph starts by describing how the authors will fit the model to the first 11 weeks of data and then project different scenarios. After that, the paragraph seems to veer into discussing historical trends. This is a bit confusing and unclear presentation. Also are weeks 11 and 12 in 2021 and data from lockdowns used to model interventions starting in week 12 of 2021? The presentation could be a bit more clear.

The model is fitted to data of the full pandemic history in the region, starting from early 2020 up to week 11 2021 (March 15-21, 2021). At the end of that week, interventions were applied to the region. For this reason, we constructed lockdown scenarios of different stringency/duration/adherence starting from week 12 2021. We revised the main text to make these elements clearer.

7. I assume the model was initialized with a population to match the area being studied, but I cannot see this specified anywhere (apologies if I missed it) and what data was used to initialize the population.

The model was initialized with an age-stratified population according to demographic and age-profile data of the region under study. Data were extracted from the French Institute of Statistics (*insee.fr*) and refer to the population at the start of year 2020. We now include the source of the data in the Methods section.

Minor

1. The paper would benefit from careful editing-there are some grammatical errors and statements that do not make much sense.

Done.

2. Line 526. I think $H_{\text{pred}}(t)$ be $H_{\text{pred}}(t|\Theta)$, else likelihood does not make much sense.

H_{pred} does indeed depend on Θ . We updated the formula to explicitly show this dependency.

3. Helpful to explain how week 11 and 12 correspond to actual dates.

Dates are now reported in the text.

Referee #2:

In this paper Di Domenico and colleagues study a range of potential scenarios for interventions to control the “third wave” of COVID-19 in Paris, France, which I believe roughly corresponds to the time period of spring and summer 2021. Mostly the scenarios include different combinations of generalized social distancing (which correspond to different degrees of maximal reductions in contacts, different amounts of loss of adherence to these policies over time, and different total durations of these “lockdowns”) and different rates of vaccine administration. They compare the number of expected hospitalizations due to COVID-19 in each scenario. In some analyses they compare this benefit of interventions with a measure of their cost to society, using a “distress” metric they invent that incorporates both the strength and duration of contact reductions. The scenarios they consider are motivated by the degree of contact reduction seen during lockdowns imposed for the “first” (spring 2020) and “second” (fall 2020) waves of COVID-19 in France.

I assume these authors are involved in very applied modeling efforts to regularly advise the French government on policy options using very data-driven/region-specific models, and that these analyses were originally conducted for that purpose. This paper seems to be the result of rebranding those analyses to say something more general about the cost/benefit trade-off between stronger but shorter interventions compared to those that are less stringent but last longer.

I think the strengths of this paper are

- * that the authors use a well-established and previously published model of COVID-19 transmission (their own) which is carefully parameterized based on data from the literature
- * the authors use a well-defined model fitting procedure which only estimates a single parameter per time period
- * the modeled impact of interventions on transmission rates is informed by real data on past interventions
- * when modeling the implementation of social distancing, the authors take into account waning of adherence to these interventions over time, again based on real data from early in the COVID-19 pandemic
- * the study attempts to quantify the social cost of interventions
- * the consideration of scenarios is very thorough: the authors considers many possible combinations of different sorts of interventions of different strengths, instead of just varying one at a time

As the authors claim in their title and abstract, their results do indeed support the idea that stricter short-term lockdowns are more effective than long-term partial restrictions.

Compared to other work that has come to the same conclusion, this paper has the benefit of using real data on the impact of policies on mobility and how they wane over time to inform their conclusions. Though as the authors themselves admit, it is difficult to know how general these findings are, since many aspects of the model are very specific to the geography and time period they consider. I do not have any concerns about the technical validity or the statistical methods used in the paper. Overall I think the analyses are solid and the results interesting.

We are grateful to the reviewer for the careful evaluation of our work, for highlighting its strengths, and for providing us with suggestions to improve the presentation of results.

The main weaknesses of the paper were in the organization and presentation of the results. I found the paper quite hard to follow. Initially, I was quite confused about what results came from modeling vs from data, and which precise period was used for model calibration vs model projections. For the first half of the paper, I think things could be clarified a lot by subdividing it further into sections and being careful not to jump around too much between topics.

We thank the reviewer for the suggestion. We revised the main text to make these elements clearer, as also suggested by the Referee #1.

I also have a few minor comments about confusing aspects of the figures:

* Abstract/Introduction: Early on and throughout the paper the authors should be more clear about the time periods being considered. When does model fitting start and end? (2020? 2021?). When does forecasting begin and end? I think it is especially confusing because there seems to be overlap in the time periods you present data for versus those you are projecting. For example, Figure 1 goes beyond week 12 with data for 2021 but then I think week 12 is when you start projecting? If possible, I would suggest using real dates, with day/month/year, not weeks, which are quite unintuitive and it is often unclear which year you are referring to. Not everyone will be familiar with when the “first wave”, “second wave” and “third wave” occurred in Paris, since waves were out of sync around the world.

We thank the reviewer for the comment, we have rephrased the first part of the Results section in order to better explain the fit/projection distinction. The model is fitted to data of the full pandemic history in the region, starting from early 2020 up to week 11 2021 (March 15-21, 2021). At the end of that week, interventions were applied to the region. For this reason, we constructed lockdown scenarios of different stringency/duration/adherence starting from week 12, 2021.

We updated the top row of **Figure 1** so that data used for inference and data outside the inference window are clearly recognisable. We report the data outside the inference window to show the full third wave. We added reference to the year and to the pandemic wave in each panel.

* There were a few words I had not heard before: performant, perduring
We corrected this throughout the main text and SI.

* Figure 1 - it's confusing that the x axis on the bottom row does not line up with the top two rows

Yes, but unfortunately data on social distancing, fear and mental health are not collected weekly, contrary to hospitalization data or mobility data shown in the first two rows (also available at daily resolution). In addition, while the interest in the first two rows was specific to each pandemic wave (a relatively short time window), a longer timeframe is necessary to observe behavioral trends. This is the reason why the third row is not aligned with the first two in terms of x axis (also, the last panel shows a correlation).

* Figure 2 - it's really hard to see differences in colors. The legend suggests there are 6 different scenarios but in most plots I only see only 4 or 5 lines. And what is the curfew scenario? (I don't think this was ever explained)

In this figure, some scenarios overlap. This is the case of short moderate lockdowns (2 weeks), as the loss of adherence is not visible, or of strict lockdown scenarios, regardless of their duration. That is why the 6 scenarios are not always visible in each plot. We added this explanation in the caption of the figure.

The curfew scenario corresponds to the projected trends based on unchanged curfew conditions as estimated in week 11, 2021. This is now defined in the text.

* Figure 3 - Is the shading of the plot markers really necessary? It is hard to see, and the x axis location gives the adherence pattern. I found these plots very hard to read with so much going on. It was hard to interpret the meaning of the symbol shape/color.

Color shades of the markers refer to the duration of the lockdown intervention, they are hence required to codify this. However, we simplified the notation of **Figure 3**, by removing the color code on the level of adherence, which is already encoded in the x axis - thanks for the suggestion.

* Figure 5 - The plots in the lowest row seem totally separate from those in the upper 2 rows ... maybe they should be a separate figure? Also the legends for the bottom row are confusing. It's really hard to tell differences between 2 line colors and what they represent. The labels on the figure didn't say this, and I had to dig through caption to figure it out

We thank the reviewer for the suggestion, however we preferred to keep the third row of plots within **Figure 5**, as this figure is meant to show the conditions reached at the exit of restrictions (first two rows, depending on vaccination rollout and seasonality) and how different hypotheses on the phasing out of these restrictions will impact the epidemic dynamics in the weeks after (third row). To simplify its readability, the color code and the notation have been changed. The legend has been modified accordingly.

REVIEWERS' COMMENTS:

Reviewer #1 (Remarks to the Author):

The authors have done a great job clearing up my questions and this paper is a lot more clear now. I think that the results are well-substantiated and the model with nicely calibrated. I find the results compelling and thoughtful.